# *MdWRKY120* Enhance Apple Susceptibility to *Alternaria alternata*

**DOI:** 10.3390/plants11233389

**Published:** 2022-12-05

**Authors:** Lifu Liu, Xiaoming Li, Wei Guo, Jiajun Shi, Wenjun Chen, Yingying Lei, Yue Ma, Hongyan Dai

**Affiliations:** 1College of Horticulture, Shenyang Agricultural University, Shenyang 110866, China; 2School of Horticulture, Liaoning Vocational College of Ecological Engineering, Shenyang 110101, China

**Keywords:** apple, WRKY, *Alternaria alternata*, disease resistance, MdWRKY120, fungal pathogen

## Abstract

*Alternaria alternata (A. alternata*) is a common pathogen that greatly influences apples’ quantity and quality. However, chemical treatments produce increased health risks along with decreased food and environmental safety. Advancements in plant molecular biology, such as transgenic technology, have increased apple trees’ resistance to pathogens and have therefore attracted widespread attention. WRKY transcription factors are involved in abiotic and biotic stress regulation; however, their biological role in non-model plants such as apple, is still unknown. In this investigation, *MdWRKY120* was isolated from the ‘GL-3′ apple to determine its function during *Alternaria alternate* infection. The MdWRKY120-GFP fusion protein was located in the nucleus. MdWRKY120 in yeast cells exhibited activating transcriptional activity, meaning it is a transcription activator. MdWRKY120 overexpression transgenic plants were more sensitive to *A. alternata*, while RNAi transgenic plants showed increased resistance to *A. alternata*. This investigation demonstrates that MdWRKY120 enhances the susceptibility of apples to *A. alternata*.

## 1. Introduction

Apple (*Malus × domestica*) is among the most important fruit crops, belongs to the family Rosaceae, and is widely cultivated globally due to its elevated nutritional and economic values. However, various fungal diseases seriously threaten the yield and quality of apples all around the world [1]. Apple *Alternaria* leaf spot is a fungal disease caused by the *Alternaria alternata (A. alternata)* and is one of the most serious diseases that affect apple [2]. Circular black or brownish leaf spots cause defoliation and reduce photosynthesis, fruit quality, and apple production [3]. Currently, chemical treatments are the main methods to control the disease; however, these treatments produce food safety concerns and environmental pollution. Sustainable crop production and minimal chemical use against plant-pathogen interactions have attracted great attention. Although extensive breeding studies have been conducted, the knowledge of genetic understanding of pathogen defense is still limited. Further, since apples have a long breeding cycle, little progress has been made in this field. Therefore, it is important to find disease-resistant genes and explore the potential molecular mechanisms associated with their anti-infection activity.

The *WRKY* family is an important transcription factor family in higher plants, regulating growth and development [4]. The WRKY transcription factor contains at least 1~60 amino acids-long conserved domain of WRKY, a highly conserved N-terminal heptapeptide called WRKYGQK, and a C-terminal-, C_2_H_2_-, or C_2_HC-type zinc finger structure [5]. Based on the number of WRKY conserved domains and the type of zinc finger structure, the WRKY transcription factors are classified into three groups [6]. Genome sequencing has helped identify many WRKY gene families in various crops. *Arabidopsis thaliana* consists of 75 genes [7], the apple of 127 genes [8], maize of 119 genes [9], and rice of 103 genes [10]. The WRKY family is crucial for regulating biotic and abiotic stress, such as drought [11], salt stress [12], disease resistance [13], seed germination [14], and flower development [15].

As has been recently reported, WRKY40s are essential for plant immune response. WRKY protein’s function against biotic stresses in horticultural plant species has also been studied. In pepper immunity, WRKY40 directly regulates *ChiIV3* expression level, thereby significantly increasing its resistance to *Ralstonia solanacearum* [16]. It was also found that, in the resistant (WR315) plants, there was WRKY40 up-regulation under Fusarium stress, whereas, in susceptible (JG62) plants, it lacked expression. Furthermore, in *Arabidopsis*, heterologous chickpea WRKY40 over-expression stimulated an immune response against *Pseudomonas syringae* [17]. In *Arabidopsis*, simultaneous *WRKY18* and *WRKY40* mutations render otherwise susceptible wild-type species resistant to the *Golovinomyces orontii,* a biotrophic powdery mildew fungus [18]. In some species, WRKY40 is reported to have opposite regulations; for instance, the GhWRKY40 transcription levels in cotton increased after bacterial pathogen *R. solanacearum* infection, and it improved most of the defense-associated genes, increased the wounding tolerance, and the resisitance to *R. solanacearum* [19]. Overall, WRKY40 significantly regulates a variety of plant stress responses; however, little is known about its function in apples against *A. alternata*.

In our previous study, we found that AtWRKY40 and MdWRKY120 are homologous genes and, thus, it was hypothesized that they have similar functions. Therefore, this study is focused on MdWRKY120, which was expressed in all the studied tissues and was relatively high in the fruit and root. The green fluorescent protein (GFP) labeled MdWRKY120 protein was localized in the nucleus and displayed transcriptional activity in yeast cells. When non-transformed plants were compared with overly expressed MdWRKY120 plants, the latter were more susceptible to *A. alternata*. Overall, MdWRKY120 is crucial for regulating *A. alternata* resistance in apples. This study on MdWRKY120 against *A. alternata* resistance will pave the foundation for the molecular genetic breeding of apple disease resistance.

## 2. Results

### 2.1. Identification of MdWRKY120 in Apple

The apple’s transcription factor MdWRKY120 gene sequence was downloaded from the JGI database (https://phytozome-next.jgi.doe.gov/ accessed on 2 November 2022). To investigate the evolutionary association between MdWRKY120 and other WRKY proteins, a phylogenetic evaluation was carried out of MEGA 6.06 software by the Neighbor-Joining method. As indicated by Figure 1a, MdWRKY120 is closely related to PbWRKY120 in pear. The sequence alignment of MdWRKY120 amino acid and other WRKY120 proteins revealed one shared conversed WRKY domain (Figure 1b). Since the WRKY120 transition factor is involved in pathogen defense and stress response, its close homology in various species may indicate some similar functions, especially in *A. alternate* defense, and some of these proteins are reported to take part in plantss defense against bacteria [17] and fungi [18].

### 2.2. Expression of MdWRKY120 in Multiple Organs

For elucidating MdWRKY120 expression levels in various tissues, real-time PCR (qPCR) was performed using four-year-old ‘GL-3’ trees. The results indicated that MdWRKY120 was expressed in different tissues, including flower, stem, leaf, root, and fruit, with the highest levels in the fruit (Figure 2).

To determine the mechanism responsible for MdWRKY120′s expression patterns, its promoter region of 1591 bp was evaluated. The Plant CARE database predicted various response elements for biotic and abiotic stresses, such as hormone responses, tissue-specific expression, light, and development, as shown in Appendix A. The core W-box, GARE *cis*-acting elements were found in the MdWRKY120 promoter region. The data collected from the above experiments suggested that MdWRKY120 may be involved in defense and stress.

### 2.3. Subcellular Localization of MdWRKY120

It was reported that most WRKYs were nucleus-localized proteins [20]. To elucidate whether MdWRKY120 was a nuclear protein like other WRKYs, the MdWRKY120 coding sequence was attached to the pRI-GFP via the CaMV 35S promoter, followed by transient expression of the MdWRKY120-GFP fusion gene in tobacco leaves. As indicated by Figure 3, the MdWRKY120-GFP fusion protein’s signals were limited to the nucleus, whereas the positive control GFP signals were expressed throughout the cell, proving that MdWRKY120 was localized in the nucleus and is thus a nuclear protein.

### 2.4. Transcription Activation Analysis of MdWRKY120

To analyze the transcriptional activity of MdWRKY120, the complete MdWRKY120 coding sequence was inserted into pGBT9 to obtain the pGBT9-MdWRKY120 construct, which was then transformed in the yeast cells. The empty pGBT9 vector was kept as the negative control. As Figure 4 indicates, the Y2H Gold yeast strain containing pGBT9 only propagated in SD/-Trp, while the yeast strain containing pGBT9-MdWRKY120 propagated in both SD/-Trp and SD/-Trp-His-Ade with X-α-Gal. The result indicated that MdWRKY120 was a transcription activator.

### 2.5. Overexpression and RNAi Knockdown of MdWRKY120 in ‘GL-3′ Apple

Furthermore, to investigate the extent to which its role is important, especially MdWRKY120’s response to *A. alternata* in apples, overly expressed ‘GL-3′ apples were developed by transforming with the pRI101-MdWRKY120 plasmid. Putative transgenic plants were acquired via *Agrobacterium*-mediated ‘GL-3′ apples’ transformation, performed following the instructions of Zhang et al. [21]. Six independent kanamycin-resistant MdWRKY120-overexpressing apple plants were propagated; its overexpression was subsequently determined by PCR analysis using a primer set MdWRKY120F/MdWRKY120R. A 963 bp band from the genomic DNA of 3 overly-expressed *MdWRKY120* transgenic lines was amplified. No corresponding bands were observed in ‘GL-3′ plants (Figure 5b). Compared with non-transformed control plants, the *MdWRKY120* expressions in the 3 lines were notably enhanced. The relative *MdWRKY120* expressions were remarkably enhanced in OE1, and OE1 -4 and -5 (Figure 5c). Thus, the *MdWRKY120* was successfully overexpressed in apples with high levels and OE1 and OE5 were selected for further investigation. ‘GL-3′ apples’ MdWRKY120 RNAi silencing was also carried out via *Agrobacterium*-mediated transformation. Three independent kanamycin-resistant plants were propagated and authenticated by PCR (Figure 5e) and qRT-PCR (Figure 5f). The MdWRKY120 expression in these propagated plants was sharply decreased, and RNAi-2 and RNAi-5 were chosen to represent MdWRKY120-RNAi transgenic plants.

### 2.6. MdWRKY120 Confers A. Alternata Resistance to Apple

In *A. alternate*-infected apples, there was subsequent cell death. The circular black or brownish spots on the leaves expanded in late spring to early summer, eventually causing defoliation. To figure out the role of MdWRKY120 in defending against *A. alternata,* the leaves 5 days post fungi inoculation were observed and the morbidity and disease index was analyzed. As shown in Figure 6a, after 2 days of fungi inoculation, brown spots appeared in MdWRKY120-RNAi plants and controls, while brown spots were much more obvious in overexpressed plants. After 3 days of inoculation, the overexpressed plants showed more disease spots than MdWRKY120-RNAi plants. In gene-silenced plants, no disease spots appeared until 5 days after inoculation. Moreover, the area of disease spots in MdWRKY120-silenced plants was markedly smaller than that in overexpression and control plants. In brief, after 5 days of inoculation, the morbidity in MdWRKY120-RNAi plants was <45%, compared with >80% morbidity of MdWRKY120 overexpressed plants and 65% of control plants (Figure 6b). The statistics of disease index after 4 days of inoculation in overexpressed plants was lower and, in the gene-silenced plants, was higher than the control. The result indicated that *MdWRKY120* enhances apple susceptibility to spotted leaf litter pathogen.

## 3. Discussion

*A. alternata* is a fungus that leads to crop diseases, especially in Asian countries such as China. Alternaria infection produces circular black or brownish spots that develop on apple leaves during late spring to early summer and which cause defoliation. The infection leads to reduced photosynthesis, fruit quality, and apple production [3]. Currently, effective treatment methods mainly include chemical use, which can cause cross-resistance [22]. The disease can also arouse the inherent immune response of plants, although little is known about this [23].

The literature suggests the involvement of WRKY proteins in microbe-associated molecular pattern-triggered immunity and other important mechanisms. The immune system of plants has two models, including (1) PAMP-triggered immunity, and (2) effector-triggered immunity. Pathogen resistance is a complicated process involving transcriptional regulation, and transcription factors are essential for plant defense responses [24,25,26]. With the recent rapid advancements in molecular biology and transgenic technology, plants’ immunity can be improved by genetic modification. WRKY transcription factors directly or indirectly regulate the expression of downstream stress resistance genes (such as disease resistance-related genes, PR) and improve pathogenic resistance by regulating the immune response in many species [27,28,29,30,31,32,33]. In a preliminary study, Hawthorns infected by apple chlorotic leaf spot virus (ACLSV) were investigated by high-throughput sequencing, and it was found that plant-pathogen interaction-related genes (WRKY transcription factors) were significantly up-regulated in the infected group [34].

WRKY40 (WRKY family member) is associated with resistance against biotic stress [35,36,37,38,39]. Many complex and overlapping networks, including WRKYs, regulate multiple biological processes in plants. The heterologous expression of GhWRKY40 enhanced the susceptibility of tobacco to *R. solanacearum* and its reduction in the expression of PR1 and PR2 genes in the SA signal transduction pathway, ACS6 genes in the ET signal transduction pathway, and JAZ1 and JAZ3 genes in the JA signal transduction pathway has been reported [29]. In poplar plants, PtrWRKY40 overexpression enhances its sensitivity to ulcerative pathogens and decreases the expression of genes related to SA signaling pathways PR1.1, PR2.1, PR5.9, and CPR5 [39]. These results suggest that WRKY40 proteins are the transcription factors that are functionally conserved and regulate plants’ abiotic stress response. According to phylogenetic alignment, MdWRKY120 and AtWRKY40 are homologous genes. MdWRKY120 protein had activating transcriptional activity and its overexpression reduced the resistance to *A. alternata*. Unlike other species, WRKY40 family genes play a negative regulatory role in the immune resistance mechanism against spotted fall disease in apples. An in-depth investigation to evaluate the exact mechanism is needed. This research puts forth crucial information regarding the apple plants’ resistance to Alternaria.

## 4. Materials and Methods

### 4.1. Plant Materials and Growth Conditions

Apple genotype ‘GL-3′ tissues were cultured in the greenhouse of Shenyang Agricultural University, China, and then used in the genetic transformations and fungal inoculation experiments. The tissue for *Agrobacterium*-mediated leaf transformation was cultured in the subculture MS medium augmented with 0.3 mg/L 6-BA, 0.2 mg/L IAA, and 0.1 mg/L GA_3_. The experiment was performed after plants grew 5–6 leaves. Tobacco (*Nicotiana benthamiana*) plants were propagated in a growth chamber at 25 °C under a 14 h day/light cycle. These were subsequently utilized for subcellular localization.

For gene expression analysis, roots, leaves, stems, fruits, and flowers of ‘GL-3′ were harvested, snap freezed in liquid nitrogen, and then stored at −80 °C.

### 4.2. Cloning MdWRKY120 Gene

The primer pairs were designed with the sequence of MdWRKY120 (MDP0000794439) and then amplified by utilizing cDNA as a template. ‘GL-3′ leaves’ total RNA was obtained via an improved CTAB protocol [24], followed by cDNA synthesis using the PrimeScript RT Reagent Kit (TaKaRa, Dalian, China). Then, with this cDNA, the full-length MdWRKY120 coding sequence was amplified. The amplification program was as follows: one cycle of 10 min at 95 °C followed by 40 cycles of 10 s at 95 °C and 30 s at 60 °C. The purification of PCR products was carried out with TaKaRa MiniBEST Agarose Gel DNA Extraction Kit Ver.4.0 (TaKaRa, Dalian, China), followed by cloning into a pMD18-T Vector (TaKaRa, Dalian, China), and, finally, sequencing.

### 4.3. Sequence Alignment and Phylogenetic Analysis

The WRKY120s amino acid sequences in different species were acquired from the NCBI based on their reported accession number and assessed via the DNAMAN 6.0 software. With the help of MEGA 6.0 software, a phylogenetic tree was generated. (PbWRKY120, XP_009349633.1; MhWRKY120, AGG23550.1; MbWRKY120, TQD95607.1; PmWRKY120, XP_008228795.1; PaWRKY120, XP_021821871.1; RcWRKY120, XP_024182559.1; FvWRKY120, XP_004303242.1; ZjWRKY120, XP_015869638.1; MeWRKY120, XP_021609717.1; AtWRKY120, NP_178199.1; SiWRKY120, XP_011073326.1; CaWRKY120, XP_027121557.1; CsWRKY120, AFJ20666.1; ZmWRKY120, AQK69076.1; OsWRKY120, AAL78375.1.)

### 4.4. Subcellular Localization

The MdWRKY120 coding sequence (in the absence of stop codon) was attached to the Green Fluorescent Protein gene, and then the fusion gene was cloned in the pRI-GFP vector under the cauliflower mosaic virus (CaMV) 35S promoter regulation. The fusion vector was introduced into *Agrobacterium tumefaciens* strain EHA105. In the 1-month-old *N. benthamiana* leaves, *A. tumefaciens* cells harboring pRI101-GFP and pRI101-MdWRKY120-GFP were administered. The GFP signals were evaluated by a confocal fluorescence microscope (Leica DMi8 A, Wetzlar, Germany) after 2 days.

### 4.5. Transcriptional Activation Analysis in Yeast Cells

The MdWRKY120 coding region was incorporated in the pGBT9 vector. All the primers that were utilized in this investigation are listed in Appendix A. The constructs of negative pGBT9-control and pGBT9-MdWRKY120 were transformed into a Y2H Gold yeast strain by following Yeast Protocols Handbook. The transformants were cultured at 28 °C on SD/-Trp for 48 h in an incubator, followed by their 10-, 100-, 1000-, and 10,000-times dilution and transfer onto the SD/-Trp and SD/-Trp/-His/-Ade plates.

### 4.6. Vector Construction and Apple ‘GL-3′ Transformation

MdWRKY120 with *Sal* I and *Kpn* I restriction sites was incorporated into the pRI101-AN vector and then sequenced at Sangon Biotech (China). The MdWRKY120 gene RNAi vector was constructed as described previously. Briefly, a 311 bp MdWRKY120 specificity fragment was incorporated into the right (*Kpn* I and *Sac* I) and left (*Xba* I and *Sal* I) pRNAi-E vector’s multiple clone sites. The primer pairs are listed in Appendix A. The prepared plasmids were then inserted into *Agrobacterium tumefaciens* strain EHA105 via the freeze-thaw method [40]. Then, the *Agrobacterium*-mediated method was performed for transforming ‘GL-3′ apples and obtaining transgenic plants [41].

### 4.7. PCR Confirmation of Transgenic Plants

For further confirmation of the positive transgenic plants, the putative transgenic and ‘GL-3′ plantlets’ total genomic DNA was obtained via a modified CTAB method [42], followed by PCR assessment with *MdWRKY120*-F/*MdWRKY120*-R primer set (Appendix A). The primer set’s amplicon size was 963 bp and the amplification program was one cycle of 5 min at 95 °C, followed by 35 cycles of 35 s at 94 °C, 30 s at 58 °C, 90 s at 72 °C, and a final 10 min extension at 72 °C. Non-transformed plant DNA was set as the negative control.

### 4.8. Gene Expression Analysis

Gene expression level was evaluated using qRT-PCR and quantified using the 2^−ΔΔCT^ method [43]. Initially, cDNA was prepared via PrimeScript RT Reagent Kit (TaKaRa, Dalian, China), followed by qRT-PCR using SYBR Premix Ex TaqII (TaKaRa, Dalian, China) on Applied Biosystems 7500 Real-Time PCR System (Applied Biosystems, Foster City, CA, USA) [42]. Apple actin gene was taken as a reference. Primers were enlisted in Appendix A. The entire experiment was performed three times.

### 4.9. Alternaria Mali Culture and Infection

The *A. alternata* fungi were propagated for 7 days on potato dextrose agar at 25 °C. Then, the ‘GL-3′ apple leaves and transgenic apple plants were both inoculated with *A. alternata* and cultured for 25 days at 23 °C in an incubator. The plants were observed daily and photographed at 0, 1, and 2 days after inoculation. After the specified period, the obtained tissue samples were snap frozen with liquid nitrogen and stored at −80 °C. Each sample was processed in three biological and technical replicates.

According to the classification standard of apple leaf spots disease:(1)
$$\mathrm{Incidence}\;\mathrm{of}\;a\;\mathrm{disease}=\frac{\mathrm{Number}\;\mathrm{of}\;\mathrm{diseased}\;\mathrm{leaves}}{\mathrm{Investigate}\;\mathrm{the}\;\mathrm{total}\;\mathrm{number}\;\mathrm{of}\;\mathrm{leaves}}\times 100\%$$
(2)
$$\begin{array}{l} \mathrm{Disease}\;\mathrm{index}\\ =\frac{\sum \left(\mathrm{Number}\;\mathrm{of}\;\mathrm{diseased}\;\mathrm{leaves}\;\times \;\mathrm{Representative}\;\mathrm{value}\;\mathrm{at}\;\mathrm{all}\;\mathrm{levels}\right)}{\mathrm{Investigate}\;\mathrm{the}\;\mathrm{total}\;\mathrm{number}\;\mathrm{of}\;\mathrm{leaves}\;\times \;\mathrm{highest}\;\mathrm{level}\;\mathrm{representative}\;\mathrm{value}}\\ \times 100\% \end{array}$$

The grading standard is shown in Table 1.

### 4.10. Statistical Analysis

SPSS 22.0 (SPSS Inc., Chicago, IL, USA) was utilized for the statistical assessment of the data. One-way analysis of variance (ANOVA) was carried out by Duncan’s multiple-range test. *p* < 0.05 was deemed a significant difference.

## 5. Conclusions

In summary, our study finds that MdWRKY120 overexpression transgenic plants were more sensitive to *A. alternata*, while RNAi transgenic plants showed increased resistance to *A. alternata*. This research puts forth crucial information regarding the apple plants’ resistance to Alternaria.

## Figures and Tables

**Figure 1 plants-11-03389-f001:**
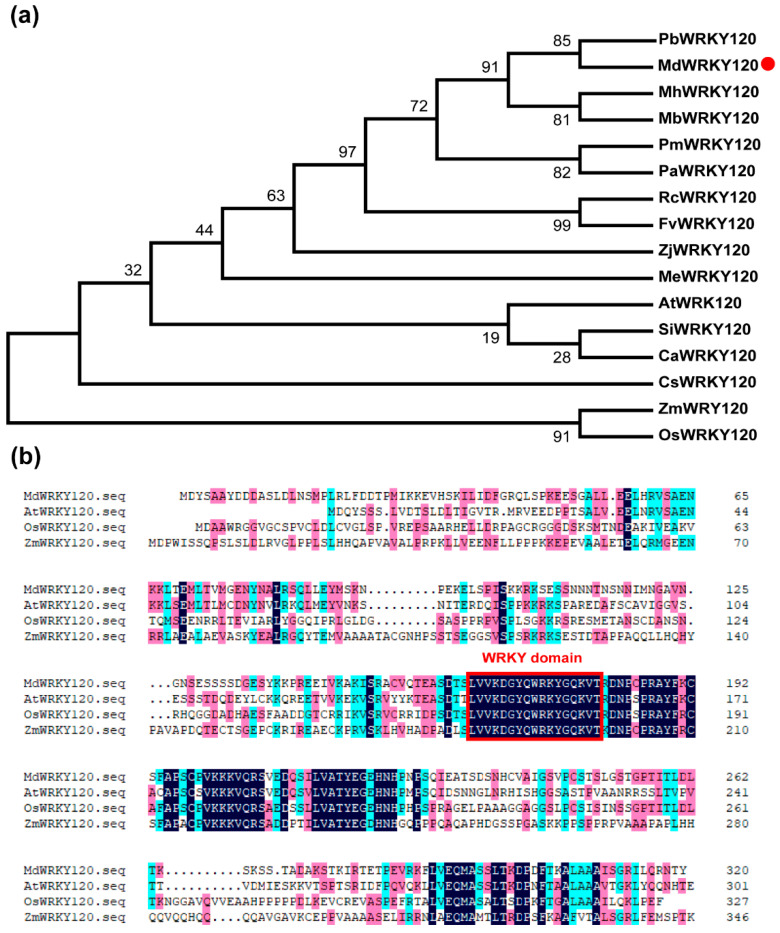
Phylogenetic alignment of MdWRKY120 homologous proteins. (**a**) The Phylogenetic tree of MdWRKY120. The red dot represents apple WRKY120. Pb, *Pyrus bretschneideri*; Md, *Malus domestica*; Mh, *Malus hupehensis*; Mb, *Malus baccata*; Pm, *Prunus mume*; Pa, *Prunus armeniaca*; Rc, *Ricinus communis*; Fv, *Fragaria vesca*; Zj, *Ziziphus jujube*; Me, *Manihot esculenta*; At, *Arabidopsis thaliana*; Si, *Setaria italica*; Ca, *Coffea arabica*; Cs, *Camellia sinensis*; Zm, *Zea mays*; Os, *Oryza sativa*. (**b**) Sequence alignment of MdWRKY120 amino acids.

**Figure 2 plants-11-03389-f002:**
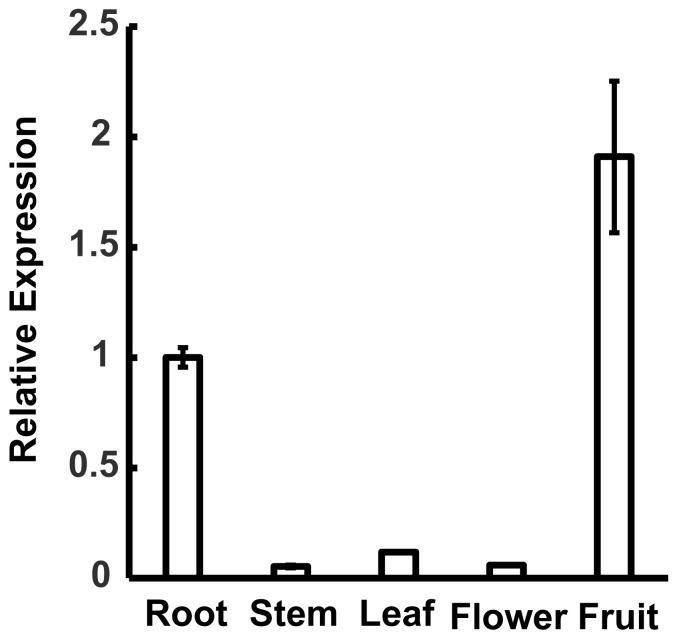
Expression patterns of MdWRKY120 in various tissues as assessed by qRT-PCR. The error bars represent data from 3 experimental replicates.

**Figure 3 plants-11-03389-f003:**
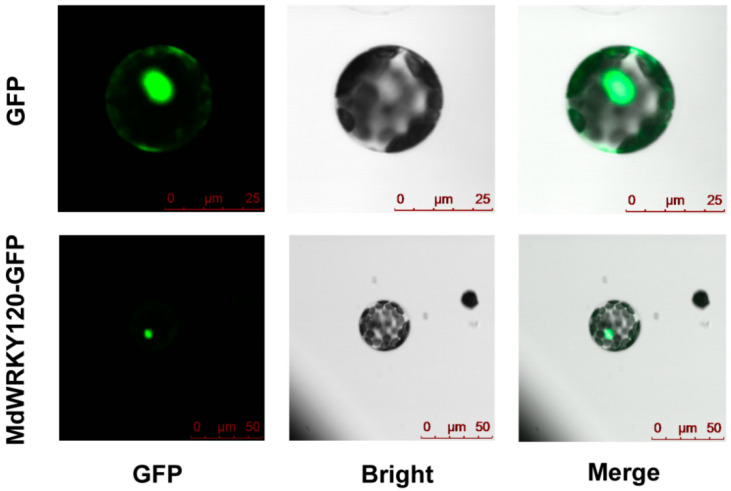
Subcellular localization of MdWRKY120. The MdWRKY120-GFP (**bottom**) and GFP (**top**) vectors were transformed in tobacco protoplasts, respectively.

**Figure 4 plants-11-03389-f004:**
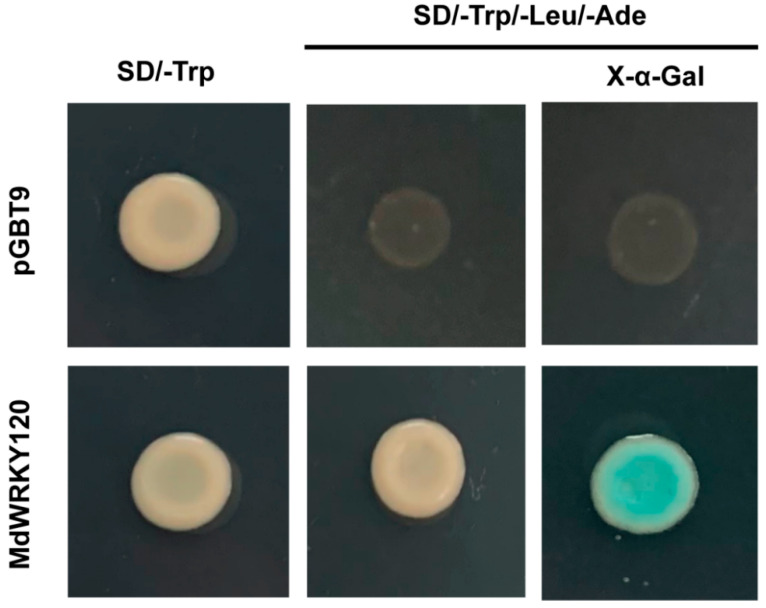
Transcriptional Activity Analysis of MdWRKY120. The empty BD (pGBT9) or yeast strains transformed BD-MdWRKY120 constructs propagated in SD/-Trp/-His/-Ade and SD/-Trp media, respectively.

**Figure 5 plants-11-03389-f005:**
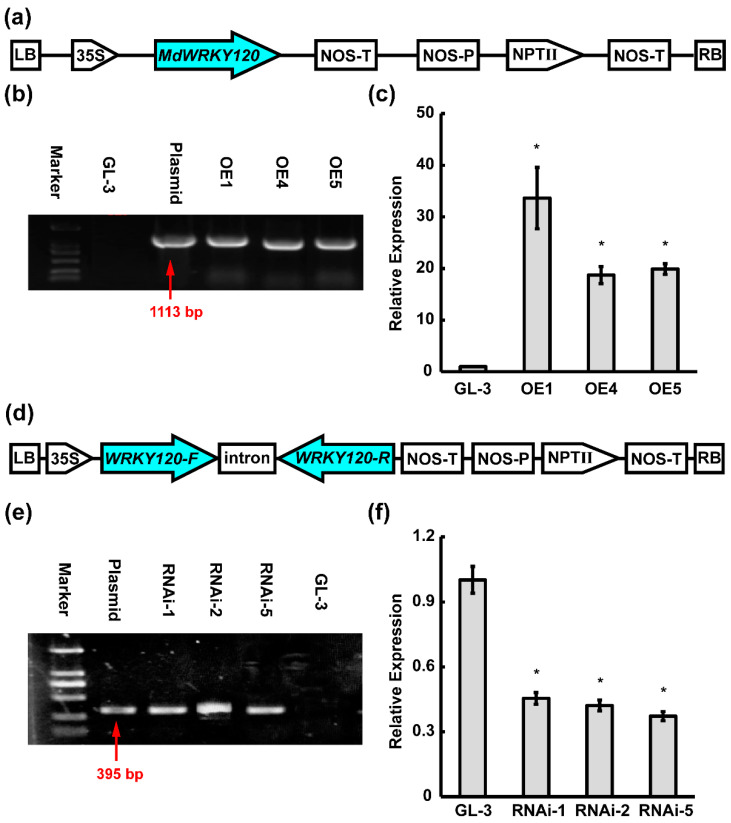
Molecular confirmation of *MdWRKY120*-overexpressing transgenic lines. (**a**) Structural diagram of the pRI101-MdWRKY120 construct with increased expression of MdWRKY120, driven by the CaMV 35S promoter. (**b**) PCR evaluation of T-DNA sequences from ‘GL-3′ plants and MdWRKY120 transgenic lines. (**c**) qPCR assessment of ‘GL-3′ plants and independent positive transgenic lines’ MdWRKY120 transcripts. Vertical bars depict the SDs (*n* = 3). ‘*’ demonstrate *p* < 0.05. (**d**) Structural diagram of the pRNAi-MdWRKY120 construct. (**e**) PCR evaluation of T-DNA sequences from ‘GL-3′ plants and *MdWRKY120* transgenic lines. (**f**) qPCR assessment of ‘GL-3′ plants and independent positive transgenic lines’ MdWRKY120 transcripts. Vertical bars depict the SDs (*n* = 3). ‘*’ demonstrate *p* < 0.05.

**Figure 6 plants-11-03389-f006:**
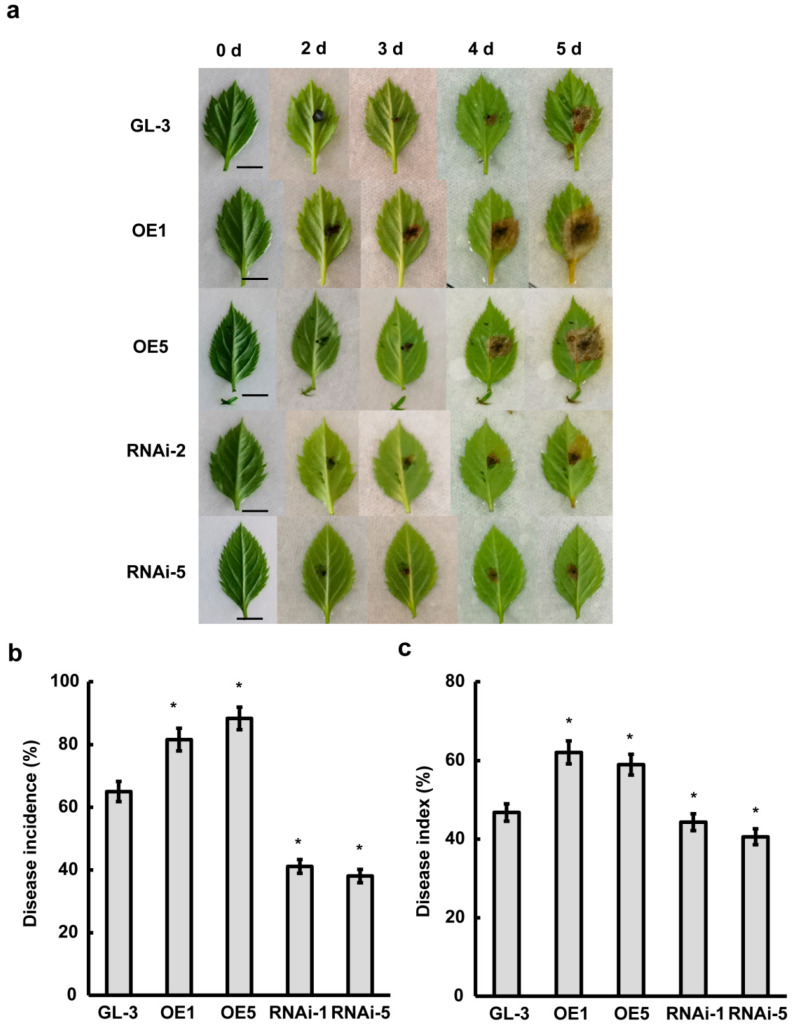
MdWRKY120 overexpression increases the sensitivity to Alternaria blotch in apples. (**a**) Disease reaction phenotypes from ‘GL-3′ and transgenic apple leaves, bars = 1 cm. (**b**) The disease incidence of apple transgenic plants and controls under *A. alternate* stress. (**c**) The disease index of apple transgenic plants and controls under *A. alternate* stress. Vertical bars depict the SDs (*n* = 3). ‘*’ demonstrate *p* < 0.05.

**Table 1 plants-11-03389-t001:** Grading standard of disease.

(X) Value of Grade	Percentage of the Disease Spots in Total Leaves	Severity
0	X = 0	No disease spot
1	0 < X ≤ 10	The disease spot area < 1/10
3	10 < X ≤ 25	1/10 < The disease spot area < 1/4
5	25 < X ≤ 50	1/4 < The disease spot area < 2/5
7	50 < X ≤ 75	2/5 < The disease spot area < 3/4
9	X > 75.0	The disease spot area > 3/4

## Data Availability

Not applicable.

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
