# Peer review of "MdWRKY120 Enhance Apple Susceptibility to Alternaria alternata"

_plants, 2022, doi:10.3390/plants11233389_

Round 1
Reviewer 1 Report
In the manuscript “plants-2042355”, the authors found that MdWRKY120 negatively regulated the resistance of apples to A. alternata. The findings are meaningful. The resistant allele may be used to improve the apple resistance to A. alternata through marker assisted selection in breeding.
Minors: (1)In Table 1,Grade 9 is defined as “the disease spot proportion is more than half”, but Grade 7 is more than half. (2) Please indicate the accession number in NCBI of AtWRKY120, OsWRKY120, and ZmWRKY120.
Reviewer 2 Report
The manuscript entitled "MdWRKY120 Negative Regulation Apple Resistance to Alternaria alternata' is focused on MdWRKY120, which was expressed in all the studied tissues but was the highest in the fruit and root. The green fluorescent protein (GFP) labelled MdWRKY120 protein was localized in the nucleus and displayed transcriptional activity in yeast cells. MdWRKY120 overexpressed transformed plants were more susceptible to A. alternate than non-transformed and this indicated that MdWRKY120 is crucial for regulating A. alternata resistance in apples.
This is a very well written article. Very compact, not contain unnecessary information. It uses many modern biotechnological techniques that require extensive experience and skills.
The title should be changed. I'll make a suggestion in the attached file, but I think it can be changed in several different ways. I don't know if I like the phrase 'negative regulation of resistance', but I might use 'enhance susceptibility' or 'vulnerability'.
The abstract could be edited in such a way as to further highlight the results obtained and emphasize their importance. Encourage the reader to read the entire article.
In the Introduction chapter, I miss information about A. alternata. And I would show the disease under study in the context of a global threat, not a local one. Many European countries are leading apple producers, does this disease matter in this part of the world? This is an international journal and such information would be welcome.
When it comes to Materials and methods, I have made some comments on the text in the attached file. For example, there is no citation to the CTAB method used as an RNA isolation method - typically used as a DNA isolation method. Stylistically wrong description of PCR programs. Some abbreviations are not explained. I think this can be checked throughout the text. It should also be remembered that each part of the experiment presented in this chapter should be described in such a way that it should be possible to repeat these experiments by other researchers.
I like the Results chapters and the Discussion chapter. I have marked some mistakes in the attached file.
The Conclusions chapter needs to be redrafted. Already in the first sentence, there is ambiguity. 'Our study shows that MdWRKY120 and AtWRKY40 are homologous genes'. This was found in the previous work and the results on this subject are not in the current work. The conclusion chapter is also an important part of the article that is very often read by other scientists, so it should be written with due care. All the most important achievements obtained in the work should be emphasized here.
I recommend a minor revision.

Round 2
Reviewer 1 Report
My concerns have been addressed in the revised manuscript "plants-2042355". I recommend the publication of the revised manuscript.